# Development, evaluation and application of a novel markerless motion analysis system to understand push-start technique in elite skeleton athletes

**Laurie Needham**[1,2]*, **Murray Evans**[1,3], **Darren P. Cosker**[1,3], **Steffi L. Colyer**[1,2]

**1** Centre for the Analysis of Motion, Entertainment Research and Applications, University of Bath, Bath, United Kingdom, **2** Department for Health, University of Bath, Bath, United Kingdom, **3** Department of Computer Science, University of Bath, Bath, United Kingdom

* ln424@bath.ac.uk

**Data Availability Statement:** All data from this project relating to athlete performances, e.g. outputs from our proposed system and ground-truth data, are embargoed at the request of our

## Abstract

This study describes the development, evaluation and application of a computer vision and deep learning system capable of capturing sprinting and skeleton push start step characteristics and mass centre velocities (sled and athlete). Movement data were captured concurrently by a marker-based motion capture system and a custom markerless system. High levels of agreement were found between systems, particularly for spatial based variables (step length error 0.001 ± 0.012 m) while errors for temporal variables (ground contact time and flight time) were on average within ± 1.5 frames of the criterion measures. Comparisons of sprinting and pushing revealed decreased mass centre velocities as a result of pushing the sled but step characteristics were comparable to sprinting when aligned as a function of step velocity. There were large asymmetries between the inside and outside leg during pushing (e.g. 0.22 m mean step length asymmetry) which were not present during sprinting (0.01 m step length asymmetry). The observed asymmetries suggested that force production capabilities during ground contact were compromised for the outside leg. The computer vision based methods tested in this research provide a viable alternative to marker-based motion capture systems. Furthermore, they can be deployed into challenging, real world environments to non-invasively capture data where traditional approaches are infeasible.

## Introduction

The skeleton push start requires the athlete to accelerate a sled to a high velocity before loading onto it to adopt a prone driving position. A fast start is considered important to overall success [1, 2] where contributions to successful skeleton push start performance include attaining a high pre-load velocity and executing an effective loading phase [3]. Physical attributes, such as lower limb power and sprinting ability, explain a large portion of the variance in the sled velocity attained [4] and as such the prescription of regular sprint training is often used as physical

research partners the British Bobsleigh and Skeleton Association (BBSA). The BBSA have requested this data be restricted at the moment, as releasing such data prior to the 2022 Winter Olympic Games could give rival teams the opportunity to analyze the biomechanical techniques of British athletes and as such allow them to gain an unfair advantage. Further restrictions are applied by the University of Bath's Ethics Committee as data sharing was only permitted between the research team and the BBSA. The University of Bath's Ethics Committee will consider data requests sent to health-ethics@bath.ac.uk.

**Funding:** This research was funded by CAMERA, the RCUK Centre for the Analysis of Motion, Entertainment Research and Applications, EP/M023281/1 and EP/T014865/1 and in collaboration with the British Bobsleigh and Skeleton Association who we thank for their continued time and support with this project. There was no additional external funding received for this study.

**Competing interests:** The authors have declared that no competing interests exist.

preparation modality. However, Colyer et al. [4] also noted that factors related to technique are likely to account for some of the remaining variance in performance.

Improved sprint times in elite skeleton athletes have been associated with increases in loading distance and pre-load velocity [5]. However, the enhancement of sprinting ability did not necessarily guarantee a faster start as reductions in loading effectiveness have been observed under higher velocity loading conditions [5]. It appears, therefore, that while sprint training has the potential to enhance push start performance, the relationship between these two activities is a complex one and pushing should not simply be viewed as bent over sprinting. Comparisons of kinematics between training activities and their target skill has demonstrated that similar movement patterns can facilitate task specific adaptations [6] and via transfer of training, enhanced performance of the target skill [1, 7]. Therefore, capturing the underlying kinematics of the athlete during pushing and sprinting is required to further understand the transfer of sprinting to pushing a skeleton sled. Such information would allow for the comparison of pushing kinematics to the more researched activity of sprinting and provide practitioners with a conceptual understanding to guide the selection of skeleton specific training activities [8].

To date, limited push start kinematic information is available, perhaps due in part to the limitations associated with current motion capture technology and the challenging environment that skeleton takes place in. For example, dynamic outdoor lighting conditions are challenging to control and account for during motion capture with marker-based systems. Furthermore, placement of reflective markers is both time consuming and invasive with the potential to alter technique and thus reduce ecological validity. Advances in computer vision and deep learning are revolutionising all aspects of movement sciences, providing viable alternatives to the traditional marker-based motion capture technologies that are considered the gold standard for many biomechanical research applications [9]. Computer vision and deep learning based approaches have advantages over marker-based motion capture and inertial measurement units (IMUs) in that they are capable of fully automating the capture and extraction of information from an image or sequence of images in a manner that is completely non-invasive to the participant. Furthermore, by capturing and storing regular images data can easily be reprocessed with the latest algorithms for improved results in this rapidly developing field. Deep convolutional neural networks (CNN) have been highly influential in the computer vision research and application, excelling at tasks such as segmentation (e.g. segmenting the outline of a person in an image) and key-point estimation (e.g. estimation of joint centre coordinates). While such methods are computationally intensive and require large amounts of training data, they consistently outperform more traditional methods [10] such as the use of binary image operations or shallow, fully connected neural networks.

Previous work from our research group presented a computer vision based multi-camera system capable of non-invasively capturing accurate step characteristic data during overground running [11]. Validation against force plates and manual camera-based analysis showed temporal variables to fall within 1.5 frames of the criterion data while spatial variables demonstrated errors of < 1 cm. Needham et al. [12] provided modifications to this system incorporating deep learning based methods in order to robustly capture step characteristic data during the skeleton push start at an outdoor training facility. Additionally, the proposed system included athlete and sled mass centre velocities averaged across each step. Evans et al. [13], further enhanced the tracking of athlete and sled providing accurate velocities as a function of time. In the work presented here, we build upon the previously proposed systems to present, validate and apply a complete motion analysis system that can capture step characteristics and mass centre behaviour for both sprinting and skeleton push starts. Furthermore, the

complete system described here offers notable performance increases over previous work. Therefore, the aim of this study was to:

1. Compare the performance of a computer vision and deep learning based system to that of a traditional marker-based motion capture system in a challenging real-world environment (skeleton push-track).

2. Utilise the proposed system to compare biomechanical indicators of performance (step characteristics and mass centre velocities) during the skeleton push start and regular acceleration sprinting.

## Materials and methods

Twelve international skeleton athletes (seven males [1.81 ± 0.05 m, 83.37 ± 2.73 kg], five females [1.71 ± 0.03 m, 70.04 ± 1.44 kg]) were recruited. All participants provided written informed consent. Ethical approval was provided by the University of Bath's research ethics committee. Approval number—EP 18/19 052. Each athlete attended two testing sessions. During session one, each athlete performed three maximal effort dry-land push starts at the University of Bath's outdoor push track. During session two, each athlete performed two 10 m sprints, two 15 m sprints and two 20 m sprints. After each pair of sprints, the starting line was moved backwards to allow different parts of the acceleration phase to be captured (Fig 1).

During both sessions, motion data were captured concurrently using two motion capture systems. Criterion data were captured using a 15-camera marker-based motion capture system (Oqus, Qualisys AB, Gothenburg, Sweden) while additional data were captured using a custom 9-camera (1920 × 1080 pixel resolution, ~90˚ field of view, JAI sp5000c, JAI ltd, Denmark) computer vision system. During session one, both camera systems were positioned around the push track in order to capture the pushing action between 5 m and 15 m from the starting block (Fig 1) where there was a constant gradient of ~2%. During session two, both cameras systems were positioned around an indoor sprints tracks in order to capture the sprinting technique between 0–10 m, 10–20 m and 20–30 m from a three-point start.

Motion capture systems were time-synchronised by means of a periodic TTL-pulse generated by the custom system's master frame grabber to achieve a frame locked sampling frequency of 200 Hz in both systems. Additionally, start and stop trigger signals for both systems

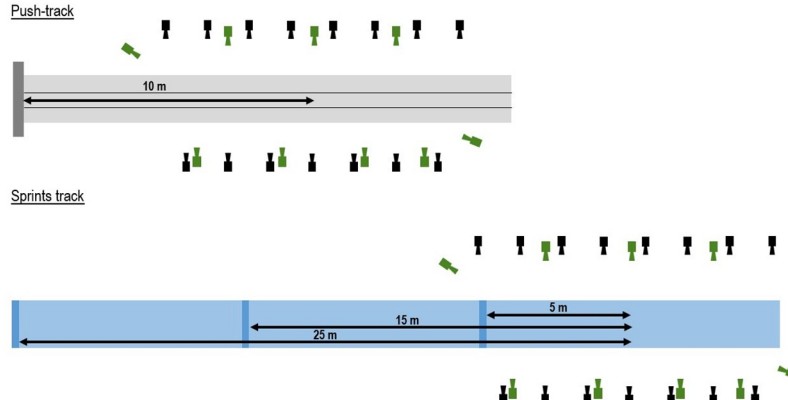

**Fig 1. Schematic of the experimental set-up at the push-track (top) and on the indoor track (bottom).** The marker-based and markerless cameras are denoted in black and green, respectively. The capture volume at the push-track was ~10 m long and was centred ~10-m from the block (red dashed area). On the indoor track, a ~10 m volume was captured across multiple trials with three different starting positions. This allowed a 0–30 m section to be reconstructed (red dashed area).

were generated by the master frame grabber on the computer vision system. This ensured that not only did both camera systems start and stop at the same time but that frames were captured by all cameras in unison without drift.

The Qualisys system was calibrated as per the manufacture's specifications. The custom camera system used a binary dot matrix to initialise each camera's intrinsic [14] before extrinsic parameters were solved via Sparse Bundle Adjustment [15] to provide a globally optimal calibration. A right-handed coordinate system was defined for both systems by placing a Qualisys L-Frame in the centre of the push track, 10 m from the start block. In order to refine the alignment of each system's Euclidean space, a single marker was moved randomly through the capture volume and tracked by both systems. This marker data provided points with which the spatial alignment could be optimised in a least-squares sense. To assess the reconstruction accuracy of both systems a wand was moved through the capture volume and tracked by both systems before the mean (± SD) resultant vector magnitude was computed and compared to the known dimensions of the wand.

To capture criterion data a full body marker set comprising of 44 individual markers and four clusters were attached to each participant to create a full body six degrees of freedom (6DoF) model (bilateral feet, shanks and thighs, pelvis and thorax, and bilateral upper arms lower arms, and hands). Four additional markers were placed on the sled to its track position and orientation. Following labelling and gap filling of trajectories (Qualisys Track Manager v2019.3, Qualisys, Gothenburg, Sweden) data were exported to Visual 3D (v6, C-Motion Inc, Germantown, USA) where raw trajectories were low-pass filtered (Butterworth 4$^{th}$ order, cut-off 12 Hz) and a 6DoF inverse kinematics (IK) constrained model was computed. Athlete mass centres (CoM) were computed using the model described by de Leva [16]. Additionally, the sled was modelled as a rigid object with uniformly distributed mass. Filtered marker trajectories and mass centre locations were exported to a custom Python script (v3.7, Python Software Foundation, USA) where mass centre derivatives were computed using a finite central differences method and touch-down (TD) and toe-off (TO) events were computed according to the method described by Handsaker et al. [17]. Computing TD and TO events permitted the calculation of step characteristics including step length (SL), step frequency (SF), step velocity (SV), step time (ST), ground contact time (GCT) and flight time (FT).

This research builds upon the computer vision based foot contact detection algorithms presented by Evans et al. [11] and Needham et al. [12]. Challenging lighting conditions at the push-track caused severe problems for typical background subtraction methods, preventing robust segmentations of the athlete (foreground mask) from the background. As such, traditional background subtraction was replaced with the CDCL-human-part-segmentation [18] and DensePose [19] which both implement a convolutional neural network (CNN) to detect and segment body parts in an image (S1 Fig in S1 File). This approach proved robust to dynamic lighting conditions and had the added advantage of not segmenting the sled as foreground. To detect approximate footfall locations the scene was divided into a horizontal grid of cells, and the occupancy of each cell at various heights calculated by accumulating the proportion of the projection that is foreground in each camera view. Peaks in the ground plane occupancy map larger than a threshold represent cells that are highly likely to contain the on-ground foot. Peaks at knee and body height can be used to verify those ground-plane peaks. Approximate timing and location of contact events can be determined by monitoring occupancy over time. Setting the ground plane to be 0.025 m above the actual ground helped to avoid partial occlusions created by the sled (S2 and S3 Figs in S1 File).

Foot position was further refined by initialising an approximately foot-sized 3D bounding box along the axis representing the direction of travel. The position of the bounding box was initialised using foot contact information from the ground plane occupancy maps before the

position was optimised to fit the foot. Further refinement of TD and TO event timings were achieved by tracking the foot in individual camera views. The foot-sized bounding box was projected into each camera view and the region inside each 2D projection was split into vertical slices. A Sobel filter was used to compute the vertical gradient of each slice, and then a horizontal mean reduces each slice to two 1D arrays of colour and gradient values. Frame to frame tracking was performed for each slice by finding the vertical offset that minimises the difference between frames. Tracking begins at the frame where the contact has the largest area in the ground plane occupancy map, typically around mid-stance. TD and TO event timings are determined by tracking from that point forwards or backwards in time to find the frame when the last slice that displaces starts to initiate vertical movement. Ascertaining TD and TO locations and timings permitted the computation of step characteristics (SL, SF, GCT, ST and FT).

Athlete CoM locations, as well as sled CoM locations during pushing, were determined using two separate methods. The first was presented by [11]. This method determines athlete CoM locations by first using a CNN [18] (e.g. DensePose) to segment the torso and head in each 2D image. The CNNs used in these experiments were pre-trained on generic datasets. 2D bounding boxes are created for each head/torso region in each camera view which found to provide an reliable proxy for the true CoM location. An occupancy map is used for initial 3D detection and cross-camera fusion. After initialisation the 3D bounding box for head/torso is optimised for size and location by projecting into the images and optimising the overlap of the projection with 2D segmentations areas. In subsequent frames, the bounding box is initialised in its previous location and re-optimised for the new segmentation, while the occupancy map verifies its continued existence and checks for new detections (S4 Fig in S1 File). Finally, the centroid of each 3D bounding box is passed through a bi-directional Kalman filter to provide an optimal state estimation and the weighted average of both body parts computed. Sled CoM was determined by using a CNN to detect the four corners of the sled (S5 Fig in S1 File) in each 2D camera view. Detections with a high confidence were back projected into the 3D space using the camera calibration and their intersect used to drive the motion of the sled which was treated as a rigid object. Finally, the centroid of the sled was passed through a Kalman filter. Athlete and sled CoM velocities were computed using a finite central differences method before being averaged across each step. The approach proposed by Evans et al. [13] will be referred to in this work as the '3D bounding box' method.

In order to evaluate system performance, the results from the computer vision systems were compared to the criterion system (marker-based motion capture) using linear regression and Bland-Altman analysis showing the mean difference (bias) and 95% limits of agreement (LoA). Further analysis comparing the pushing and sprinting data used estimation statistics [20], which focuses on the magnitude of the effect and its precision. Gardner-Altman estimation plots [21] were produced containing paired Cohen's $d$ effect sizes and 95% bootstrap confidence intervals (CIs). Five-thousand bootstrap samples were taken and the confidence interval was bias-corrected and accelerated [22]. $P$ values were computed using a Permutation t-test. The reported $P$ values represent the likelihood of observing the effect size, if the null hypothesis of zero difference is true. For each permutation $P$ value, 5000 reshuffles of the control and test variables were performed. Effect sizes and CIs are reported as: effect size [CI width, lower bound; upper bound]. Effects sizes were interpreted according to Cohen's [23] guidelines (small effect $\geq$ 0.2, moderate effect $\geq$ 0.5 and large effect $\geq$ 0.8).

## Results

Motion capture system mean reconstruction accuracy was 0.91 ± 0.76 mm for the criterion system and 0.74 ± 0.68 mm for the computer vision system demonstrating high accuracy for

**Table 1. Between system comparison of computed step characteristics during pushing.**

| Variable | Mean Difference (Bias) | ± SD | 95% LoA | $R^2$ |
|---|---|---|---|---|
| GCT (s) | 0.008 | 0.015 | 0.037 | 0.10 |
| FT (s) | -0.008 | 0.016 | 0.023 | 0.28 |
| ST (s) | 0.001 | 0.017 | 0.033 | 0.20 |
| SL (m) | 0.001 | 0.012 | 0.021 | 0.99 |
| SF (Hz) | -0.022 | 0.148 | 0.278 | 0.20 |

**Table 2. Comparison of computed CoM step velocities during pushing and sprinting.**

| Activity | Variable | Mean Difference (Bias) | ± SD | 95% LoA | $R^2$ |
|---|---|---|---|---|---|
| Pushing | Athlete CoM Velocity (m.s$^{-1}$) | -0.017 | 0.080 | 0.140 | 0.94 |
| Pushing | Sled Velocity (m.s$^{-1}$) | 0.015 | 0.023 | 0.061 | 0.99 |
| Sprinting | Athlete CoM Velocity (m.s$^{-1}$) | 0.004 | 0.068 | 0.138 | 1.00 |

both systems. Validation results for pushing temporal step characteristics are given in Table 1. High levels of agreement were observed for temporal variables between the proposed computer vision system and the criterion system (Table 1), with differences falling within 1.5 frames. Excellent agreement was observed for spatial variables such as SL with mean differences of 0.001 ± 0.012 m.

For pushing using the 3D bounding box based method, mean differences between the criterion and computer vision based step-averaged sled velocities were -0.015 ± 0.023 m.s$^{-1}$, while athlete CoM differences were -0.017 ± 0.080 m.s$^{-1}$ (Table 2). For sprinting using the 3D bounding box method, mean differences for the athlete step-averaged CoM were 0.004 ± 0.068 m.s$^{-1}$. Bland-Altman and linear regression plots can be found in S6–S13 Figs in S1 File.

The following results were derived using the experimental computer vision system. Step characteristics as a function of step velocity are given in Fig 2 for both pushing and sprinting.

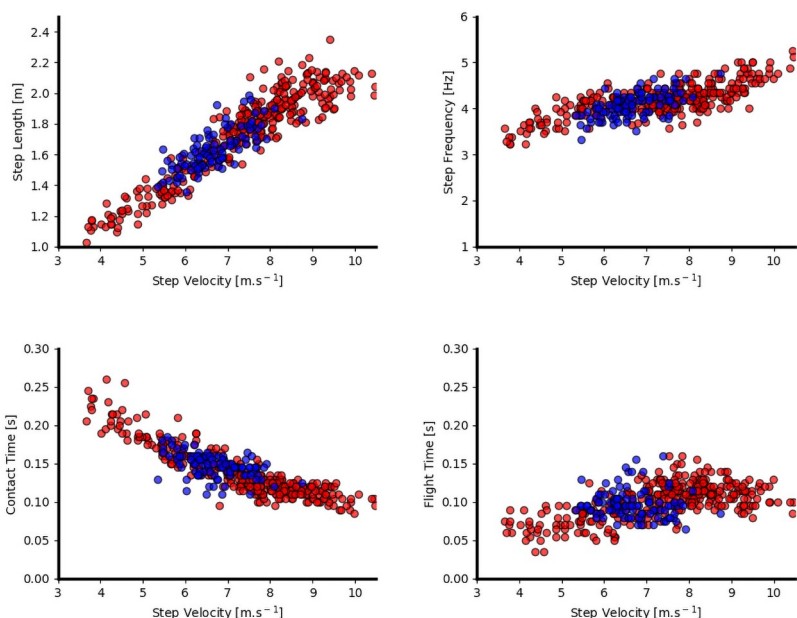

**Fig 2. Step characteristics during sprinting (red) and pushing (blue) shown as a function of step velocity.**

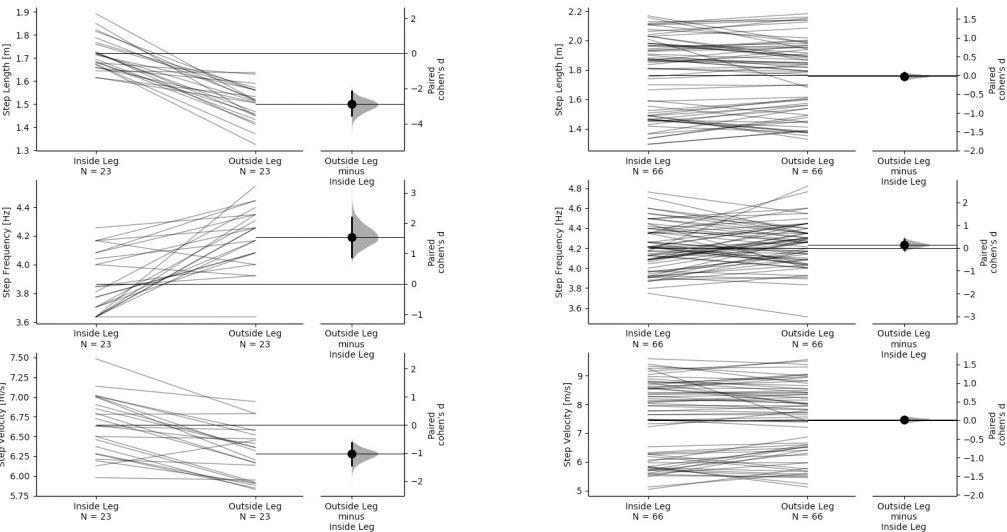

**Fig 3. Paired step characteristic differences for inside and outside leg during pushing (left column) and the corresponding leg during sprinting (right column).** The paired Cohen's d is plotted on floating axes on the right as a bootstrap sampling distribution. The Cohen's d effect size is depicted as a dot; the 95% confidence interval is indicated by the ends of the vertical error bar. Top row—SL, centre row—SF, bottom row—SV.

Step characteristics inter-limb asymmetries for both pushing and sprinting are given in Figs 3 and 4 as estimation plots showing the paired differences. Results show that when pushing, athletes exhibit similar step characteristics to those observed during sprinting at similar velocity. The results of the estimation statistics including confidence upper and lower bounds and P values are given in S1 Table in S1 File.

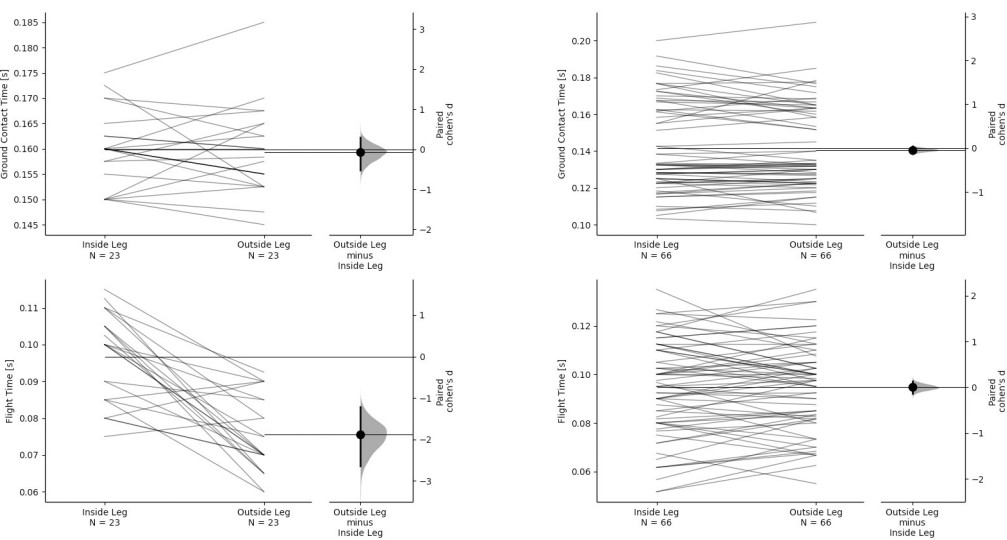

**Fig 4. Paired step characteristic differences for inside and outside leg during pushing (left column) and the corresponding leg during sprinting (right column).** The paired Cohen's d is plotted on floating axes on the right as a bootstrap sampling distribution. The Cohen's d effect size is depicted as a dot; the 95% confidence interval is indicated by the ends of the vertical error bar. Top row—GCT, bottom row—FT.

## Discussion

The ability to non-invasively capture skeleton push-starts and sprinting kinematics in challenging environments would allow coaches to better understand the key aspects within and between these two training activities. In this study a computer vision and deep learning methodology was developed and validated, with its utility to accurately extract biomechanical characteristics and allow novel, previously-inaccessible information to be captured in a challenging real-word environment demonstrated.

Accuracy of spatial variables such as SL was excellent with a mean difference between systems of 0.001 m. This particular outcome was not surprising as the methods used to calibrate both camera systems and reconstruct 3D locations were fundamentally the same. However, this is of course dependent on each system's ability to reliably detect and track a location on the foot and demonstrates that this can indeed be achieved in the absence of markers. Temporal variables such as GCT and FT presented higher differences (Table 1) but these were still within approximately 1.5 frames of the criterion system. Due to occlusion caused by the sled, the ground plane with which footfall events were detected was raised by 0.025 m which caused a systematic overestimation of GCT and systematic underestimation of FT as a result of early touch-down detection and late toe-off detection. However, mean differences were similar to those reported for the widely used OptoJump™ system (Microgate, Bolzano, Italy) (0.005 ± 0.004 s, [24]) which also has a raised ground plane. Furthermore, ST (GCT + FT) differences were lower (0.001 ± 0.017 s) as the errors from overestimated GCT and underestimated FT effectively cancel each other out, further indicating that errors in temporal variables were likely due to the elevated ground plane. Alternative approaches using wearable technology such as body or foot mounted accelerometers [25] or in-shoe pressure measurement [26] also offer measurement of temporal but not spatial step characteristics. Mean GCT errors of 0.0017 s have been reported for trunk mounted accelerometers [27] and errors of −-0.0067 ± 0.0229 s for in-shoe pressure measurement [28] which fall within the same range of measurement error as the vision based system used in this study. However, both wearable approaches are highly sensitive to changes in walking/running velocity, are untested in the skeleton environment and are ultimately more invasive in nature than a vision-based approach.

Sled velocities and athlete mass centre velocities (Table 2) demonstrated excellent agreement with the marker-based system where mean step-averaged differences of -0.017 m.s$^{-1}$ for the athlete CoM and -0.015 m.s$^{-1}$ for the sled were observed. Notably in this work, we demonstrate athlete and sled tracking performance increased upon our previous work [12, 13] with substantial reductions mean difference SDs and substantial increases in R$^2$ values (Table 2). The performance increases were attributed to more robust outlier detection and improved noise process noise predictions in the Kalman filtering stage. Furthermore, the computer vision methods demonstrated superior performance to other field-based methods for measuring athlete running velocities such as laser distance measurement, which exhibits mean errors of up to 0.41 ± 0.18 m.s$^{-1}$ [29], two to four times higher than the results of this study. Within the application of this study (skeleton push start and sprint accelerations), it appears that the proposed deep learning based computer vision methods provide an alternative, non-invasive way to capture important technique related information (step characteristics and mass centre velocities) where more conventional systems would not be viable. More work is required to reduce measurement errors of some temporal step characteristics (e.g. FT and GCT during pushing) in the proposed markerless methods before they are comparable to marker-based motion capture. However, this field of research is advancing rapidly, and such improvements are likely to emerge in the near future.

Sled pushing step characteristics as a function of distance demonstrated a decrease in SL, SF and FT, and increase in GCT as a result of the external load created by the sled and the bent over postures required to push the sled. These findings align with other studies examining the effects of external resistance (15–30% of body weight) on sprinting performance [28, 30–32] but such studies only considered the overall differences between sprinting with and without external resistance and did not examine how step characteristics between the two activities align as a function of step velocity. Comparison of GCT and FT as a function of step velocity revealed similar results between pushing and sprinting (Fig 2) despite the clear differences in the constraints of these movements. This suggests that sprinting activities at similar velocities to those achieved during pushing broadly mimic these high-level biomechanical variables (step characteristics). Exposure to low resistance sprints training such at sled pushing has been demonstrated to increase the athlete's ability to develop horizontal forces at higher velocities [33], a training stimulus that the highly trained athletes of this study have been exposed to in large volumes.

The pushing data which were captured between 5 m and 10 m from the starting line, aligned with sprinting data captured between 0 m and 10 m for all athletes suggesting that there may be a greater level of biomechanical specificity earlier in the sprinting phase where postures, velocities and step characteristics more closely match those of pushing. Therefore, it may be more appropriate to match the two activities as a function of velocity and not simply as a function of distance when considering how sprinting activities could be used to replicate pushing. Brazil et al. [34], encouraged coaches to consider the principle of specificity in a holistic way considering all the biomechanical information that is available to ensure that a) the most appropriate training activities are selected and b) the likelihood of achieving positive training adaptation and enhancing performance of the target skill is maximised. Consequently, coaches are now able to make evidence-based decisions to meet the criteria described by Brazil et al. [34], by using the system detailed in this study.

Analysis of the athlete velocities during pushing between approximately 5 m and 15 m and athlete sprinting velocities between 0 m and 10 m further demonstrates that when pushing, the athlete achieves comparable velocities later in the acceleration phase. This is perhaps unsurprising given that during pushing the athlete is highly constrained and must accelerate the sled from a static position. Previously, Colyer et al. [35], has demonstrated very strong associations between push-start performance and both 15–30 m unresisted sprint time and 0–15 m resisted sprint time (sled load, 10 kg for males and 7.5 kg for females). This observation supports the notion of matching sprinting and push activities using CoM velocity and further demonstrates that early acceleration sprints training may be mechanically more specific to pushing, at least for the phase of pushing being analysed in this study.

Comparison of inside leg and outside step characteristics revealed significant asymmetries and large effect sizes during pushing for SL, SF, SV and FT but not for GCT (S1 Table in S1 File). No significant asymmetries were observed for any step characteristic variable during sprinting. The large effect sizes and significant differences observed during pushing are likely the result of the asymmetrical pushing technique where the athlete must push the sled with one arm while the contra-lateral arm is free to swing. Athletes exhibited longer step lengths but lower step frequencies on the inside leg, which typically resulted in higher step velocities. The higher step frequencies on the outside leg were due to reduced flight times as there were no clear differences between GCTs. It is worth noting that the athletes never actually lose full contact with the ground due to the requirement to push and remain in contact with the sled using the support arm. However, athletes appear to spend a similar amount of time imparting an impulse in foot-ground contact, regardless of which leg, inside or outside, is in contact with the ground. During upright sprinting, a lower flight time and step length would indicate lower

force production during the preceding contact [36, 37]. The observed asymmetries could, therefore, indicate a compromise in force production capabilities during the outside leg contact. However, this asymmetry could also reflect differences in how effectively energy is transferred to the sled. When generating force on the outside leg, due to the greater distance from the sled to the point of force application on the ground, the transfer of energy to accelerate the sled could conceivably be less efficient than when contacting the ground with the inside leg. Alternatively, it could be that in the bent-over position, athletes are "pulled" (due to the requirement to maintain contact with the sled) towards the ground during flight more on the outside than the inside leg, which would also reduce flight time and step length. A full body kinetic and kinematic analysis of the pushing technique is required to further understand how pushing a sled alters acceleration mechanics and creates the asymmetries observed in this study.

## Conclusions

A novel computer vision and deep learning based approach to non-invasively capture kinematic data was thoroughly validated for skeleton push starts. The method was applied in a challenging real-world environment and application (skeleton push starts and sprinting) and was able to capture representative kinematic data including step characteristics and mass centre velocities of the athlete and sled through a fully automated end-to-end workflow. This approach could be employed by coaches and practitioners to monitor technique where traditional motion capture techniques are not practical. The computer vision based system was used to capture skeleton push start and sprinting data where step characteristics were found to be comparable between activities when matched as a function of step velocity. The developed system allowed large asymmetries to be observed between the inside and outside leg during pushing that were not present during sprinting. However, regardless of the leg that is in contact with the ground, inside or outside, the time spent pushing against the ground applying force was not different, potentially indicating compromised outside-leg force production that should be considered by skeleton coaches. These performance observations demonstrate the utility of this system to monitor skeleton athletes' progress in response to training.

## Supporting information

**S1 File.**
(PDF)

## Author Contributions

**Conceptualization:** Laurie Needham, Murray Evans, Darren P. Cosker, Steffi L. Colyer.

**Data curation:** Laurie Needham, Murray Evans.

**Formal analysis:** Laurie Needham, Murray Evans.

**Funding acquisition:** Darren P. Cosker.

**Investigation:** Laurie Needham, Murray Evans, Darren P. Cosker, Steffi L. Colyer.

**Methodology:** Laurie Needham, Murray Evans, Steffi L. Colyer.

**Project administration:** Laurie Needham, Darren P. Cosker, Steffi L. Colyer.

**Resources:** Laurie Needham.

**Software:** Laurie Needham, Murray Evans.

**Supervision:** Darren P. Cosker, Steffi L. Colyer.

**Validation:** Laurie Needham, Murray Evans.

**Visualization:** Laurie Needham, Murray Evans.

**Writing – original draft:** Laurie Needham.

**Writing – review & editing:** Laurie Needham, Murray Evans, Darren P. Cosker, Steffi L. Colyer.

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
