## [Decision Letter · Decision Letter 0]

26 Apr 2021

PONE-D-20-34441

Development, Evaluation and Application of a Novel Markerless Motion Analysis System to Understand Push-Start Technique in Elite Skeleton Athletes

PLOS ONE

Dear Dr. Needham,

Thank you for submitting your manuscript to PLOS ONE. After careful consideration, we feel that it has merit but does not fully meet PLOS ONE’s publication criteria as it currently stands. Therefore, we invite you to submit a revised version of the manuscript that addresses the points raised during the review process.

Make sure to address all the comments provided by the reviewers, in particular you should clearly highlight your contributions and how they extend those already published in your conference paper [33]. We would also like to remind you that you cannot not reuse text which has already been published.

We look forward to receiving your revised manuscript.

Kind regards,

Jean-Christophe Nebel, Ph.D

Academic Editor

PLOS ONE

Journal Requirements:

This investigation was part-funded by CAMERA, the Reserch Councils UK Centre for the Analysis of Motion, Entertainment Research and Applications, EP/M023281/1.

Reviewers' comments:

Reviewer's Responses to Questions

**Comments to the Author**

1. Is the manuscript technically sound, and do the data support the conclusions?

Reviewer #1: Yes

Reviewer #2: Partly

Reviewer #3: Yes

2. Has the statistical analysis been performed appropriately and rigorously? 

Reviewer #1: Yes

Reviewer #2: No

Reviewer #3: Yes

3. Have the authors made all data underlying the findings in their manuscript fully available?

Reviewer #1: Yes

Reviewer #2: Yes

Reviewer #3: No

4. Is the manuscript presented in an intelligible fashion and written in standard English?

Reviewer #1: Yes

Reviewer #2: Yes

Reviewer #3: Yes

5. Review Comments to the Author

Reviewer #1: This paper deals with a very interesting application of recent computer vision and deep learning developments. It shows how those development can be applied to real-world problems. It is also very interesting to see the accuracy of those developments in comparison to marker-based methods.

My main concern with this paper is related to the similarity with a previous paper from the same authors:

[33] Needham, L., Evans, M., Cosker, D., & Colyer, S. (2020, March). Using Computer Vision and Deep Learning Methods to Capture Skeleton Push Start Performance. In 38th International Conference on Biomechanics in Sport.

The authors must highlight the differences with their previous work. Some parts of this paper are very similar to [33], particularly Sections "Materials and methods", "Results" and "Discussion". This is most of the paper.

My experience with DensePose and, probably, CPDP (I don't have experience with this model) is that the segmentation is not accurate. The segmentation misses some points in the border of the body. This can be seen in Fig. S1, it seems that the segmented feet are smaller than in the real image. However, this doesn't seem to have an effect in the results presented in this paper. This is probably to the fact that the ground plane is elevated. However, the authors mention that this elevation avoids occlusions with the sled, but I think that this elevation what favours the most is the issue of segmenting smaller feet. In the future this could be solved either with better segmentation methods or by applying a 3D mathematical morphology operation to "recover" those missed points in the border of the body.

- Line 203: It is not clear to me how to read Fig S3 when the authors say "occupancy maps shown on the left side" in the caption.

- Line 210: which colour and gradient based features are computed

Why do you think that the differences between both systems are higher for SF in Table 1?

Line 326: Table S2 is not in the Supplementary material

Line 335: I cannot find a numerical value in the paper or the supplementary material that supports that FT and GCT during pushing are the measures that require a higher reduction of measurement errors.

Line 376: As Table S2 is missing, I don't know if the reference to Table S1 is correct here and, then, I am reading the correct information

Minor comments:

- Line 84: However, in a previous study...

- Line 281: What is SV?

Reviewer #2: This paper presents a study on utilizing some existing computer vision techniques to capture sprinting and skeleton push start step characteristics and mass center velocities.

It is more like a project report rather than a scientific paper. I cannot find its originality at a research paper level. For example, it utilized CDCL-human-part-segmentation [28] and DensePose [21] for detecting and segmenting body parts, and utilized a method in [17] for athlete CoM and sled CoM localization.

It is not suitable for publishing in its present form.

Reviewer #3: In this paper, the authors present and evaluate a deep-learning-based stereovision system in the context of elite skeleton. The obtained results are very promising for step length estimation and velocities. However, there are room for improvement for temporal variables.

The authors also present first motion analysis results which i found a bit confusing since the accuracy of the system for such dynamic action still needs to be established and some points of the implementation of the system are superficially described.

My specific comments are below.

1. The pixel resolution of captured images from the computer vision system is not specified in the text.

2. Fig 1 should be made clearer with some more annotations, for instance about starting line and direction of the athlete.

3. Some more details about calibration would be needed. What was the calibration pattern for extrinsic parameters computation ? What was the diameter of the marker for alignment of systems' coordinate systems ? What was the tracking technique for the videos ? Same question for the wand.

4. Lines 193 - 199 contain elements of discussion. The authors justify the use of a DL approach but this should be discussed later in the paper.

5. Line 199 "To detect approximate foot contact locations and timings, foreground segmentations were fused and occupancy maps of the ground plane were computed". This sentence seems to me essential to assess the originality of the approach, but the interested reader has not enough elements to fully understand it. I think the authors should develop a bit on this point.

6. The authors mentioned that the sled corners were detected thanks to a CNN. Which one ? How it was trained ?

7. Similar question about the network used for COM inference. On which database has it been trained ? I think the authors should discuss the need (or not) of collecting a specific database for such dynamic motion.

8. The triangulation method is superficially described and if I'm not mistaken the 3D inference of COM was not part of ref 28.

9. What is the accuracy of COM localization when compared to qualisys ?

10. The authors claim in the abstract that "temporal variable were within 1.5 frames of the criterion measures". This statement is wrong since it is a mean (not-signed) difference between the two system. The uncertainty is between 0.02 and 0.04. So the difference between the two systems for GCT can be 8 frames. The authors should discuss these errors in the context of an elite practice.

11. An important aspect of any motion analysis system is not treated or discussed : inter-trial variability of the computed parameters.

12. Minor comment, in table 1. The unit for SF should be Hz.

13. While the first part of the results section is well balanced between the tables and the text (good synthesis of the tables in the text), FiG2 to 4 are not synthetized at all in the text.

14. It is interesting that the authors develop on some findings on their trial and new system. But since it is a novel system, it would be appreciable to see if these results could be verified on the ground-truth system.

6. PLOS authors have the option to publish the peer review history of their article (what does this mean?). If published, this will include your full peer review and any attached files.

Reviewer #1: No

Reviewer #2: No

Reviewer #3: No

---

## [Author Response · Author response to Decision Letter 0]

11 Jun 2021

Response to Reviewer #1:

Thank you for your helpful comments. We have replied below in red.

Reviewer #1: This paper deals with a very interesting application of recent computer vision and deep learning developments. It shows how those development can be applied to real-world problems. It is also very interesting to see the accuracy of those developments in comparison to marker-based methods.

My main concern with this paper is related to the similarity with a previous paper from the same authors:

[33] Needham, L., Evans, M., Cosker, D., & Colyer, S. (2020, March). Using Computer Vision and Deep Learning Methods to Capture Skeleton Push Start Performance. In 38th International Conference on Biomechanics in Sport.

The authors must highlight the differences with their previous work. Some parts of this paper are very similar to [33], particularly Sections "Materials and methods", "Results" and "Discussion". This is most of the paper.

The conference abstract in question (presented at International Society of Biomechanics in Sports annual conference) was a pilot study and represented an early version of our vison-based system. It should also be noted that this work was not peer reviewed to the same level that a paper might be in a journal. In the current work we present a later iteration of our methods which have evolved substantially in terms of how the system works and its performance. To address these concerns, we have added the following information:

Additional technical details have been added to the methods section for clarity. Lines 200-250

In the results section you will see some important performance increases when compared to our pilot work. For instance, random error between our proposed system and criterion system for athlete CoM velocity and sled CoM velocity were reduced from 0.186 m.s−1 and 0.133 m.s−1 to 0.08 m.s−1 and 0.023 m.s−1 respectively. Such performance improvements substantially improved our systems ability to detect small, meaningful changes in push start and sprinting performance. Additionally, our validation in the present manuscript is substantially more detailed with more trials and multiple exercise modalities being included (pushing and sprinting). Furthermore, we provide full Bland-Altman analyses and linear fit models for all variables assessed, allowing the reader to make a more informed decision regarding the statistical evaluation of our work. This paper is multidisciplinary in nature and provides extensive evaluation against a gold standard motion capture method in biomechanics research. This was an essential step to demonstrate to the biomechanics research community, in particular, that markerless techniques can perform to comparable levels with established marker-based techniques. 

This in turn allows us to provide real-world application of our approach to provide novel biomechanical insight to inform Olympic coaches and athletes about skeleton pushing technique which is discussed in detail in the discussion.

My experience with DensePose and, probably, CPDP (I don't have experience with this model) is that the segmentation is not accurate. The segmentation misses some points in the border of the body. This can be seen in Fig. S1, it seems that the segmented feet are smaller than in the real image. However, this doesn't seem to have an effect in the results presented in this paper. This is probably to the fact that the ground plane is elevated. However, the authors mention that this elevation avoids occlusions with the sled, but I think that this elevation what favours the most is the issue of segmenting smaller feet. In the future this could be solved either with better segmentation methods or by applying a 3D mathematical morphology operation to "recover" those missed points in the border of the body.

We too observe that these segmentation algorithms can produce segmentations that are smaller than the actual area of the person or incomplete, however, the exact area of the segmentation does not directly matter. In the occupancy map stage, the most important thing is that there is sufficient segmentation information to identify the presence of the foot (one might say we need an accurate segmentation, but not an especially precise/detailed one). For localisation the segmentation must give enough of an indication of the volume/position of the foot to guide the optimisation of the 3D bounding box location, and there are various elements in the design of the optimisation which mitigate partial segmentations. Obviously, there are limits, and as we develop our system we are considering ways of getting better segmentations from these and other algorithms - but those developments are for future work and not part of the presented work. It should also be pointed out that the elevated plane does not affect the optimisation of the 3D bounding box that provides the precise step localisation – that still uses a bounding box assumed to be on the z=0 plane.

- Line 203: It is not clear to me how to read Fig S3 when the authors say "occupancy maps shown on the left side" in the caption.

We have re-worded the caption and additionally provided further technical detail regarding how occupancy maps are used in the methods section. Lines 200-207.

- Line 210: which colour and gradient based features are computed

A Sobel filter was used to compute the vertical gradient of each slice, and then a horizontal mean reduces each slice to two 1D arrays of colour and gradient values. Lines 213-215.

Why do you think that the differences between both systems are higher for SF in Table 1?

Step Frequency (SF) refers to the cadence or number of steps per second. In this study we computed SF as 1/ST where ST (step time) was computed as the sum the GCT and FT. As such for both marker-based and markerless methods, SF is highly sensitive to any changes in GCT or FT measurements. We believe that this high sensitivity to differences or changes in one or both measurement system may have contributed to the higher differences reported.

Line 326: Table S2 is not in the Supplementary material

This should in fact refer to Table 2 in the results section. Typo corrected. 

Line 335: I cannot find a numerical value in the paper or the supplementary material that supports that FT and GCT during pushing are the measures that require a higher reduction of measurement errors.

Our mean differences (bias) between for temporal variables such as GCT and FT were low (Table 1), however, the LoA demonstrated that the range of likely values from future measurements is still higher than those seen for marker-based methods. This notion is further supported by the lower R2 values. 

Line 376: As Table S2 is missing, I don't know if the reference to Table S1 is correct here and, then, I am reading the correct information

Yes, this is referring to the correct table. Additional detail has been added to the table caption.

Minor comments:

- Line 84: However, in a previous study...

Sentence restructured for clarity.

- Line 281: What is SV?

Step velocity (SV) – definition added to methods section line 189.

Response to Reviewer #2:

Thank you for your comments. We have replied below in red.

Reviewer #2: This paper presents a study on utilizing some existing computer vision techniques to capture sprinting and skeleton push start step characteristics and mass center velocities.

It is more like a project report rather than a scientific paper. I cannot find its originality at a research paper level. For example, it utilized CDCL-human-part-segmentation [28] and DensePose [21] for detecting and segmenting body parts, and utilized a method in [17] for athlete CoM and sled CoM localization.

The present manuscript is not simply a vision based paper but rather provides a multidisciplinary approach utilising knowledge from computer vision, deep learning and sports biomechanics. We present the latest iteration of our system for non-invasively measuring sprinting and pushing technique before rigorously validating this method against the de-facto gold standard and finally applying the method to solve a real-world problem. The method presented allowed us to obtain novel results of the first biomechanics analysis of skeleton push starts and compare directly to sprinting technique in an elite population. Given all of the above, we respectfully disagree and believe that the originality of this scientific study is clear.

It is not suitable for publishing in its present form.

Response to Reviewer #3:

Thank you for your helpful comments. We have replied below in red.

Reviewer #3: In this paper, the authors present and evaluate a deep-learning-based stereovision system in the context of elite skeleton. The obtained results are very promising for step length estimation and velocities. However, there are room for improvement for temporal variables.

The authors also present first motion analysis results which i found a bit confusing since the accuracy of the system for such dynamic action still needs to be established and some points of the implementation of the system are superficially described.

If we understand your comment correctly, you have concerns over the accuracy of the marker-based system in these types of movements. Whilst we acknowledge that there is a small degree of measurement error in the marker-based system, most notably due to skin movement artifact, such limitations are well studied and can be effectively mitigated. As such marker-based motion capture systems remain the gold standard in this field and thus provide s us with a clear criterion to evaluate against. Further technical detail pertaining to the implementation of the system have been added to the methods section. Lines 165-239.

My specific comments are below.

1. The pixel resolution of captured images from the computer vision system is not specified in the text.

The JAI machine vision cameras record with an image resolution of 1920 × 1080 pixels – This has been added to the methods section. Lines 144-145.

2. Fig 1 should be made clearer with some more annotations, for instance about starting line and direction of the athlete.

Figure and caption updated to provide more information. See methods section, lines 159-162.

3. Some more details about calibration would be needed. What was the calibration pattern for extrinsic parameters computation ? What was the diameter of the marker for alignment of systems' coordinate systems ? What was the tracking technique for the videos ? Same question for the wand.

Calibration of the machine vision cameras used a calibration board with 10 x 9 78.5 mm black circles on it. Observations of the calibration board were used to initialise each camera’s intrinsic parameters (Zhang, 2000) before extrinsic parameters were initialised from pairs of cameras with shared board observations. A global optimisation was performed using Sparse Bundle Adjustment (Triggs et al., 2003) to determine the final intrinsic and extrinsic parameters. Alignment of the two camera systems coordinate systems tracked a 16 mm retro-reflective motion capture marker which was tracked manually in the machine vision system’s image data. Further detail has been added to the methods section. 

4. Lines 193 - 199 contain elements of discussion. The authors justify the use of a DL approach but this should be discussed later in the paper.

Discussion of segmentation and background subtraction techniques is outside the scope of this paper. The information provided in the methods is designed a) distinguish our method from previous approaches and b) provide the reader with some insight into our thought process during development. Therefore, we believe these do warrant the further explanation in the methods section.

5. Line 199 "To detect approximate foot contact locations and timings, foreground segmentations were fused and occupancy maps of the ground plane were computed". This sentence seems to me essential to assess the originality of the approach, but the interested reader has not enough elements to fully understand it. I think the authors should develop a bit on this point.¬¬

Further detail has been added to the methods section. Lines 199-220.

6. The authors mentioned that the sled corners were detected thanks to a CNN. Which one ? How it was trained ?

To detect the corners of the sled a keypoint detection model based on DeeperCut with a ResNet-150 backbone, was trained via transfer learning using approximately 200 annotated images. This was achieved using a modified version of the DeepLabCut framework. Further detail has been added to the methods section. Line 222-239.

7. Similar question about the network used for COM inference. On which database has it been trained ? I think the authors should discuss the need (or not) of collecting a specific database for such dynamic motion.

The segmentation networks have been taken “off the shelf”, and have been either CDCL or DensePose – they are not trained on skeleton specific datasets but have generally performed adequately, but one would clearly expect that training on skeleton and sprinting specific data would offer some advantage in segmentation quality, but how much that would improve tracking quality is less obvious. This is explained in the lines 194-200.

8. The triangulation method is superficially described and if I'm not mistaken the 3D inference of COM was not part of ref 28.

We have clarified the text of our description regarding the 3D reconstruction approach, which is fully described in ref: Line199-201 & 234 – 236.

9. What is the accuracy of COM localization when compared to qualisys ?

The fused bounding box method is not designed to provide a measure of CoM position that is directly comparable with the marker-based CoM position. The variable of interest to end users (e.g. biomechanists, coaches and athletes) is the CoM velocity. As such our method uses a weighted average of the fused torso and head bounding box centroids whose derivative provides an excellent proxy for the true CoM velocity (e.g. Table 2). Further detail regarding this has been added to the methods section (Lines 230-251). 

10. The authors claim in the abstract that "temporal variable were within 1.5 frames of the criterion measures". This statement is wrong since it is a mean (not-signed) difference between the two system. The uncertainty is between 0.02 and 0.04. So the difference between the two systems for GCT can be 8 frames. The authors should discuss these errors in the context of an elite practice.

We have reworded this sentence to acknowledge that on average, values were generally within ± 1.5 frames of the ground truth measure. This value was derived from the Bland-Altman analysis bias and as such is in fact a signed mean. Results, including the bias and random error, are discussed in relation to the performance of other measurement technologies to which we demonstrate comparable performance. It should also be noted that while such technologies are commonly used to collect running based data they cannot be used in the challenging skeleton training environments. This is discussed in lines 325-344.

11. An important aspect of any motion analysis system is not treated or discussed : inter-trial variability of the computed parameters.

Inter-trial variability of the differences between the measurement systems is presented in the Bland-Altman analyses. I.e. the SD of bias and the 95% LoA. Biological variability of the participants is outside the scope of this study but could certainly be assessed in future studies and/or taken into account when evaluating longitudinal changes in these parameters.

12. Minor comment, in table 1. The unit for SF should be Hz.

Thank you, typo corrected.

13. While the first part of the results section is well balanced between the tables and the text (good synthesis of the tables in the text), FiG2 to 4 are not synthetized at all in the text.

Further detail has been added to the results section. Lines 282 – 283.

14. It is interesting that the authors develop on some findings on their trial and new system. But since it is a novel system, it would be appreciable to see if these results could be verified on the ground-truth system.

The purpose of this paper was to present a detailed and robust evaluation of our novel system before discussing the biomechanics information provided and how it relates to both sprinting and pushing techniques. The results presented in Figs 2 – 4 are the same results that have been used to verify our novel system against the ground-truth system in Table 1 -2, Table S1 and Figs S6 – 13. Thus if we have understood your comment correctly, this verification has already been performed.

---

## [Decision Letter · Decision Letter 1]

11 Jul 2021

PONE-D-20-34441R1

Development, Evaluation and Application of a Novel Markerless Motion Analysis System to Understand Push-Start Technique in Elite Skeleton Athletes

PLOS ONE

Dear Dr. Needham,

Thank you for submitting your manuscript to PLOS ONE. After careful consideration, we feel that it has merit but does not fully meet PLOS ONE’s publication criteria as it currently stands. Therefore, we invite you to submit a revised version of the manuscript that addresses the points raised during the review process.

If you could address the remaining minor comments, we would be happy to publish your manuscript.

We look forward to receiving your revised manuscript.

Kind regards,

Jean-Christophe Nebel, Ph.D

Academic Editor

PLOS ONE

Journal Requirements:

Reviewers' comments:

Reviewer's Responses to Questions

**Comments to the Author**

1. If the authors have adequately addressed your comments raised in a previous round of review and you feel that this manuscript is now acceptable for publication, you may indicate that here to bypass the “Comments to the Author” section, enter your conflict of interest statement in the “Confidential to Editor” section, and submit your "Accept" recommendation.

Reviewer #2: (No Response)

Reviewer #3: (No Response)

2. Is the manuscript technically sound, and do the data support the conclusions?

Reviewer #2: Yes

Reviewer #3: Yes

3. Has the statistical analysis been performed appropriately and rigorously? 

Reviewer #2: Yes

Reviewer #3: Yes

4. Have the authors made all data underlying the findings in their manuscript fully available?

Reviewer #2: Yes

Reviewer #3: Yes

5. Is the manuscript presented in an intelligible fashion and written in standard English?

Reviewer #2: Yes

Reviewer #3: Yes

6. Review Comments to the Author

Reviewer #2: I have read the revised paper and the response to the reviewers. It is a good project report but not a scientific research paper.

Reviewer #3: The authors have well adressed all my comments except 2, that remained not understood by the authors but also maybe because of the lack of clarity of these 2 comments.

The 2 comments are actually linked.

Comment 1 : "The authors also present first motion analysis results which i found a bit confusing

since the accuracy of the system {i precise today i was talking about the markerless system] for such dynamic action still needs to be established"

Comment 2 : "It is interesting that the authors develop on some findings on their trial and new

system. But since it is a novel system, it would be appreciable to see if these results

could be verified on the ground-truth system."

My concern with these 2 comments is that biomechanical interpretations are given thanks to the results of the markerless system. While the validity of the system has been quantified previously against a marker-based system, would the conclusions drawn by the authors would have been exactly the same if the data coming from the marker-based system were used instead ? The assymetries that are depicted are that clear that i presume it is the case but if so, the authors could mention it.

I have a last question. Why comparison results for GCT, FT etc are given only for pushing and not on sprinting ? These variables are analyzed in fig 3 and 4 but we do not know their accuracy.

7. PLOS authors have the option to publish the peer review history of their article (what does this mean?). If published, this will include your full peer review and any attached files.

Reviewer #2: No

Reviewer #3: No

---

## [Author Response · Author response to Decision Letter 1]

14 Jul 2021

Reviewer #2: 

I have read the revised paper and the response to the reviewers. It is a good project report but not a scientific research paper. 

We respectfully disagree with this opinion. In addition to the novel scientific contributions which we outlined in our previous response, the paper in question demonstrates that computer vision based technologies can be applied to have real-world impact. This potential impact could fall across a range of science and medicine disciplines, beyond the specific application (or ‘project’) of skeleton studied here. For example, the principles demonstrated in our paper could be applied by motor rehabilitation scientists and clinicians who quantify movement to inform rehabilitation design and evaluate the effects of disease and treatments. Neuroscientists studying brain-movement interaction and motor learning. Similarly, psychologists could examine human motor development, human motor behavior and the effects of psychological disorders on movement. Sports and exercise physiologists and biomechanists could examine the metabolic costs of human movement, sports techniques (as in our paper), injury mechanisms and equipment design. Finally, engineers could quantify movement for prosthetics, exoskeleton, and rehabilitation robotics design. The scope and breadth of these examples demonstrate the substantial impact that using computer vision technologies applied to human movement research can makes to science and medicine. As such, we feel that we have provided a clear and justified rationale for the scientific contribution this paper makes and why it is suitable for publication in PLOS One. 

Reviewer #3: 

The authors have well addressed all my comments except 2, that remained not understood by the authors but also maybe because of the lack of clarity of these 2 comments. 

The 2 comments are actually linked. 

Comment 1: "The authors also present first motion analysis results which i found a bit confusing since the accuracy of the system {i precise today i was talking about the markerless system] for such dynamic action still needs to be established" 

Comment 2: "It is interesting that the authors develop on some findings on their trial and new system. But since it is a novel system, it would be appreciable to see if these results could be verified on the ground-truth system." 

My concern with these 2 comments is that biomechanical interpretations are given thanks to the results of the markerless system. While the validity of the system has been quantified previously against a marker-based system, would the conclusions drawn by the authors would have been exactly the same if the data coming from the marker-based system were used instead? The assymetries that are depicted are that clear that i presume it is the case but if so, the authors could mention it. 

Thank you for clarifying these specific comments. For clarification, any variable that has been presented in this study using our markerless system, has been verified against the ground-truth data (marker-based motion capture). Any systematic or random differences that were reported between the two measurement systems were shown to be small enough to allow for meaningful changes in push start and sprinting technique to be detected. Differences between the systems and thus the accuracy of our markerless system, in relation to the ground truth (marker-based motion capture) are presented in Tables 1 and 2 as well as in the Supporting Information. 

I have a last question. Why comparison results for GCT, FT etc are given only for pushing and not on sprinting? These variables are analyzed in fig 3 and 4 but we do not know their accuracy. 

As is discussed in the methods section, sprinting validation results for GCT, FT etc are presented in our groups previous work using an earlier iteration of the methods presented in this study. For full details see Evans M, Colyer S, Cosker D, Salo A, Ieee, editors. Foot Contact Timings and Step Length for Sprint Training. 18th IEEE Winter Conference on Applications of Computer Vision (WACV); 2018 Mar 12-15; Nv2018.

---

## [Decision Letter · Decision Letter 2]

25 Oct 2021

Development, Evaluation and Application of a Novel Markerless Motion Analysis System to Understand Push-Start Technique in Elite Skeleton Athletes

PONE-D-20-34441R2

Dear Dr. Needham,

We’re pleased to inform you that your manuscript has been judged scientifically suitable for publication and will be formally accepted for publication once it meets all outstanding technical requirements.

Kind regards,

Leonardo A. Peyré-Tartaruga, Ph.D.

Academic Editor

PLOS ONE

Additional Editor Comments (optional):

Congrats Dr. Laurie, although the reviewers have raised some points, you have been done a good job here, and the paper is ready for publishing.

Reviewers' comments:

Reviewer's Responses to Questions

**Comments to the Author**

1. If the authors have adequately addressed your comments raised in a previous round of review and you feel that this manuscript is now acceptable for publication, you may indicate that here to bypass the “Comments to the Author” section, enter your conflict of interest statement in the “Confidential to Editor” section, and submit your "Accept" recommendation.

Reviewer #3: All comments have been addressed

Reviewer #4: (No Response)

2. Is the manuscript technically sound, and do the data support the conclusions?

Reviewer #3: Yes

Reviewer #4: Yes

3. Has the statistical analysis been performed appropriately and rigorously? 

Reviewer #3: Yes

Reviewer #4: Yes

4. Have the authors made all data underlying the findings in their manuscript fully available?

Reviewer #3: (No Response)

Reviewer #4: Yes

5. Is the manuscript presented in an intelligible fashion and written in standard English?

Reviewer #3: Yes

Reviewer #4: Yes

6. Review Comments to the Author

Reviewer #3: All comments have been adressed. I recommend acceptance of this paper for publication in plos one

Reviewer #4: The authors had two aims: 1) Develop and compare a markerless motion analysis system (deep learning) with a traditional marker-based motion system performances during the tasks of skeleton push-track and sprinting. 2) Compare biomechanical variables of skeleton puh-track vs. sprinting by healthy adults using the new markerless motion analysis system.

The markerless method development is well explained, but some questions appear at Results and Discussion section.

As major point, the three paragraphs from discussion (Lines 367 – 401) discuss data not presented in the Results section. Despite the relevance from the arguments brought, I think that the Discussion thinking line should be supported by the data presented previously. Below the details of each paragraph is depict, but maybe the authors should review these arguments in overall.

The paragraph from Line 367 refers to the variation of spatiotemporal parameters of sled pushing in function of distance, but the data and figure 2 only present the variation of these parameters in function of step velocity.

The paragraphs from Lines 381 and 393 brings informations about to spatial segments from the pushing and sprinting trials, despite these specific data are not presented in Results section.

Perhaps if the authors could exhibit these informations in more details at Results (perhaps Supplementary Material), the reader’s capacity to follow the arguments would be improved.

Minor points

Line 106 – the IMU acronym definition seem unnecessary, considering that it does not appear again in the document.

Line 186 – same as above for the IK acronym.

Line 187 – reference 15 is not “de Leva”. Actually, “de Leva” is not in reference list.

Line 192 – I am in doubt if the authors analyzed a step or stride cycle from the gait: considering the step as the cycle between the same event repetition from ipsilateral to contralateral leg, while stride cycle as the cycle between the same event repetition from ipsilateral-to-ipsilateral leg (Whittle, 2007). Because if the authors refer to step cycle, it seems a bit odd to have both ground contact time and flight time from the same leg measured.

Line 201 – CNN acronym was already defined in Introduction section line 110.

Line 234 – Maybe it would increase the readability to write the authors’ name before reference 11 number, as made in Line 197.

Line 274 – It is not clear what was the authors’ criteria to consider the agreement as “excellent”.

Table 1 and 2 – I could not find the tables’ footnotes to make them self-supporting, explaining the abbreviations, per example.

Line 280 and 282 – it seems to lack the word “speed” after CoM.

Line 297 – I assume that the sentence “Results show that when (…)” refers to Figure 2. But the way that is presented - with the preceding sentence calling Figures 3 and 4 - make a misunderstanding to where to check visually the information. Perhaps explicitly calling Figure 2 in this sentence would fix this.

Figure 3 and 4 – the Coehn’s horizontal lines of zero and mean values are overlapping the left part from the figures; I think that this do not has any valid meaning. Maybe erasing this excess of line would help the figure to come clearer.

Line 325 – CV acronym was not defined before.

Line 339 – considering that ST = GCT + FT, I think that the gait cycle analyzed was stride.

Line 361 - “mass center” means CoM? Anyway, “center” is misspelling.

7. PLOS authors have the option to publish the peer review history of their article (what does this mean?). If published, this will include your full peer review and any attached files.

Reviewer #3: No

Reviewer #4: **Yes: **André Ivaniski-Mello

---

## [Editor Report · Acceptance letter]

5 Nov 2021

PONE-D-20-34441R2 

Development, Evaluation and Application of a Novel Markerless Motion Analysis System to Understand Push-Start Technique in Elite Skeleton Athletes 

Dear Dr. Needham:

I'm pleased to inform you that your manuscript has been deemed suitable for publication in PLOS ONE. Congratulations! Your manuscript is now with our production department. 

Kind regards, 

on behalf of

Professor Leonardo A. Peyré-Tartaruga 

Academic Editor

PLOS ONE